# Polyethylene Glycol: The Future of Posttraumatic Nerve Repair? Systemic Review

**DOI:** 10.3390/ijms20061478

**Published:** 2019-03-24

**Authors:** Adriana M. Paskal, Wiktor Paskal, Piotr Pietruski, Pawel K. Wlodarski

**Affiliations:** 1Laboratory of Centre for Preclinical Research, Department of Research Methodology, Medical University of Warsaw, Banacha 1B, 02-091 Warsaw, Poland; adriana.paskal@gmail.com (A.M.P.); wiktor.paskal@wum.edu.pl (W.P.); 2Timeless Plastic Surgery Clinic, gen. Romana Abrahama 18/322, 03-982 Warsaw, Poland; pietruski.piotr@gmail.com

**Keywords:** polyethylene glycol, nerve injury, treatment, review

## Abstract

Peripheral nerve injury is a common posttraumatic complication. The precise surgical repair of nerve lesion does not always guarantee satisfactory motor and sensory function recovery. Therefore, enhancement of the regeneration process is a subject of many research strategies. It is believed that polyethylene glycol (PEG) mediates axolemmal fusion, thus enabling the direct restoration of axon continuity. It also inhibits Wallerian degeneration and recovers nerve conduction. This systemic review, performed according to the Preferred Reporting Items for Systematic Reviews and Meta-Analyses (PRISMA) guidelines, describes and summarizes published studies on PEG treatment efficiency in various nerve injury types and repair techniques. Sixteen original experimental studies in animal models and one in humans were analyzed. PEG treatment superiority was reported in almost all experiments (based on favorable electrophysiological, histological, or behavioral results). To date, only one study attempted to transfer the procedure into the clinical phase. However, some technical aspects, e.g., the maximal delay between trauma and successful treatment, await determination. PEG therapy is a promising prospect that may improve the surgical treatment of peripheral nerve injuries in the clinical practice.

## 1. Introduction

Peripheral nerve injuries (PNI) remain a challenging clinical problem. Recovery after PNI is often unsatisfactory with persisting neurological deficits such as sensory and motor malfunction [1,2,3]. The outcomes depend on many factors: location, type and size of injury, time between injury and treatment, and patient’s age [4,5,6]. Most PNI result chiefly from vehicle accidents (46.4%), penetrating traumas (23.9%), and falls (10.9%) [7]. Often underestimated, iatrogenic nerve injuries constitute 17.4% of all traumatic nerve injuries, according to Kretschmer et al. [8]. A review by the same authors stated that 94% of iatrogenic nerve lesions occurred during orthopedic procedures (26%) [9]. Regardless of the cause, when nerve injury leads to axon discontinuity, a pathophysiological cascade of degeneration and subsequent regeneration of the distal nerve part occurs [10].

### 1.1. Nerve Injury Pathophysiology

Disruption of the axolemma (neuron membrane) initiates Ca^2+^ influx which leads to axolemmal sealing—leaving axon components separated proximally and distally to the injury site [11,12,13]. As a consequence of the separation of axons from the nerve cell body, the distal nerve stump undergoes Wallerian degeneration [14,15]. During this process, phagocytes clear the debris of the degenerated axons and myelin sheaths, sparing the Schwann cells, which are crucial in the regeneration process [16,17,18,19]. Experimental models and clinical examinations confirmed that the nerve regeneration rate oscillates between 1 and 3 mm/day [20,21]. The recovery time depends on the location of the PNI: a longer distance between the nerve injury and the target structures results in longer regeneration time. During that period, denervated structures, e.g., muscles and skin, are affected by the loss of innervation [22,23]. Muscles that are denervated for a long time become atrophic and eventually fibrotic [24]. Muscle fibrosis is irreversible. Muscle fibrosis would not occur only under the following conditions: when newly regenerated fibers (1) reach the target muscles, (2) produce functional synapses (neuromuscular junctions), and (3) form motor units all within 6–12 months after PNI [1]. Axon regeneration depends on the chemical communication with the Schwann cells of a distal stump [25]. Denervated Schwann cells gradually die after PNI, with the greatest reduction of their number occurring 3–6 months after PNI. This loss of Schwann cells is the main limitation in the axon regeneration process [26,27]. Unfortunately, in many cases, natural neuroregeneration occurs too late. Therefore, there is a need for new therapies accelerating that process, such as surgical interventions, pharmaceuticals, rehabilitation, stem cell and gene therapies [28,29,30,31,32,33,34,35]. For the past three decades, reports have suggested that a possible solution to avoid Wallerian degeneration is the use of polyethylene glycol (PEG) [36,37].

### 1.2. Mechanism of Polyethylene Glycol Therapy

Fusogens are chemical agents mediating cell fusion. PEG is one of the most effective fusogens known to date. PEG induces cell fusion via cell aggregation and membrane modification [38]. Specifically, PEG removes plasmalemmal-bound water, which opens the axonal ends on both sides of the injury. This enables separated axons to reconnect if they are closely apposed (cut injury). Restoration of electrical conduction, measured by compound action potentials (CAPs), indicates the effectiveness of PEG treatment [39]. Furthermore, Bamba et al. a published radiological proof of PEG-induced axonal tract reestablishment/relink through the repair site, visualized by diffusion tensor tractography of PEG-fused fixed sciatic nerves [40]. When axon ends are distantly disconnected (e.g., a crush injury), PEG mediates membrane sealing (self-fusion within the axon) (see Figure 1) [37].

The protocol of effective PEG-induced axonal fusion was invented and improved by Bittner et al. [36,41,42]. It starts with a wash in a Ca^2+^-free solution (in order to avoid Ca^2+^-mediated axolemmal sealing), then the nerve ends are coapted with epineurial microsutures. Next, washes are done with 1% methylene blue (antioxidant, for 1–2 min), 50% PEG (2–5 kDa, preferably 3.35 kDa) for 1–2 min, and finally isotonic saline containing Ca^2+^ (e.g., Ringer’s Lactate) [13,43,44,45,46,47]. Worth highlighting is the fact that protocols include no exact dose (volume) of PEG solution. The volume of solution needed to cover the whole treated nerve varies each time. An isolated PEG application, independent of the protocol, does not result in better PNI recovery [48]. Apart from the direct use of a PEG solution on the nerve coaptation site, PEG is also used in bridging nerve gaps, as a conduit material (solid PEG), or as a conduit filling (PEG solution) [49,50].

From the clinical point of view, it is worth mentioning that PEG is considered safe to use in humans. Although no study included in this systemic review state any side effects of PEG treatment, a review on its pharmaceutical applications by D’souza and Shegokar summarized that toxicity, mutagenicity, as well as teratogenic tests in rats and human clinical trials have proven its safety via oral and non-oral routes [51]. Still, PEG is often used as a drug carrier (PEGylated conjugates, surface-coating with PEG) which can lead to the formation of anti-PEG antibodies [51]. However, a single intraoperative exposition to a PEG solution causes a relatively small risk of antibodies formation compared with chronic PEGylated drug administration.

## 2. Methods

We performed a systemic review of the literature according to PRISMA guidelines. The search was carried out on 1 December, 2018, using PubMed and ScienceDirect as data sources. We searched classical articles and clinical trials. The following keywords were used: “Polyethylene glycol” OR “PEG” AND “nerve” AND “injury”. The inclusion criteria included articles which reported in vivo PEG treatment of PNI in animal models and human subjects. Non-English articles were excluded.

## 3. Results

On the basis of the search criteria, eight studies were found that met the inclusion criteria. During the reference analysis of the retrieved articles, nine additional articles were identified. Finally, 17 studies on an animal models (1 study on a guinea pig model and 15 studies on a rat model) and 1 study in human subjects were selected out of a total of 447 screened records. Excluded studies mainly comprised PEG treatments in the spinal cord injury model and applications of solid PEG as a conduit or scaffold material. Figure 2 presents a flow diagram of the literature search and selection process.

George D. Bittner is considered a pioneer of PEG therapy in PNI. His first report was published in 1985 [36]. Until 2002, all studies were performed ex vivo. Table 1 summarizes the studies on PEG treatment of PNI on animals from 2002 to 2018 (see Appendix A, a comprehensive version of Table 1 available as Appendix A). The majority of experiments was conducted on the rat sciatic nerve injury model. Different PEG application protocols were evaluated on crush and cut injuries. The results were evaluated with electrophysiological recordings, behavioral (motor) testing (e.g., Sciatic Functional Index, Foot Fault Test), histological tissue analysis, and axonal dye diffusion. In 14/16 experiments, electrophysiological recordings were carried out. Measured parameters included: CAPs, CMAPs (compound muscle action potentials), and muscle contraction force. Postrepair restoration of CAPs/CMAPs conduction through a lesion site is an indicator of successful PEG-mediated axonal fusion. On the basis of the restoration of CAPs, the first 100% success rate of PEG treatment (*n* = 32) was reported by Bittner et al. in a rat sciatic nerve cut injury model [52]. The behavioral (motor) testing is the most relevant part of neuroregeneration outcomes evaluation, as it is the closest model to a clinical setting. In most studies that implemented motor recovery analysis, PEG-treated animals showed better performance than controls in the short term, which is consistent with a rapid nerve conduction recovery caused by PEG. In long-term observations, this tendency persisted in all, except two experiments [47,53]. Histological analysis is the most objective method from those listed, as tissue harvesting is independent of animals’ generated disruptions, unlike electrophysiological recordings and behavioral testing. PEG-treated animals had a significantly higher number of axons in nerve parts distal to the injury site compared with controls (reported in five experiments, for details see Appendix A) [45,53,54,55,56]. Also, axonal diameters in the PEG-treated groups were significantly larger than the respective controls [57,58,59]. Almost all studies, except one by Brown et al., reported some kind of superiority of PEG treatments. The study by Brown et al. stated that PEG addition is not beneficial in facial nerve cut injury with suture-based repair [60]. Other dissimilarities in the study by Brown et al. are a lack of evaluation with electrophysiological recordings and a lack of nerve histological analysis. The results were based on motor function evaluation (eye blink reflex and vibrissae movement), axonal dye diffusion for motor neuron survival assessment, and histological muscle analysis. The authors implied that the facial nerve is not as susceptible as the sciatic nerve to PEG fusion. However, after taking into consideration the consistent physiology and histology within one species, that implication may be false. Bittner et al. speculated that the unusual PEG fusion protocol used by Brown et al. might not be optimal, and the lack of results of PEG fusion could be due to the use of incorrect solutions [61]. Issacs et al. also did not observe enhanced nerve regeneration (measured by muscle contraction force and nerve diameter assessment) [48]. It should be noted, however, that their study was designed mainly to evaluate whether PEG application reduced scar formation compared to fibrin glue (and indeed it proved PEG’s superiority in reducing a scar’s thickness). PEG was not applied in accordance with the protocol considering the status of Ca^2+^ during nerve repair, therefore, a lack of axon fusion disproves its effectiveness reported elsewhere.

Attempts to improve the PEG fusion protocol by supplementation with substances of supposed pro-neuroregenerative potential, such as melatonin, methylprednisolone, methylene blue, protein kinase A inhibitor (PKI), protein kinase C isozyme η pseudosubstrate fragment (ηPSF), and protein kinase C isozyme θ pseudosubstrate fragment (θPSF), were largely unsuccessful [46,52,62]. Only a methylene blue solution, when applied just before PEG treatment, significantly enhanced nerve fusion, and consequently this compound was utilized in subsequent experiments [45,53,54,55,56,57,58,59,60]. Other modifications of the fusion protocol were related to the technique of PEG application on the nerve lesion site. If specified, in nearly all protocols, the solutions were administered via a hand-held syringe. Riley et al. proposed the use of a device that minimizes the inconsistency of manual PEG solution application. This device improved the successful postoperative CAPs recovery from 72% (13/18 animals) in the standard PEG application method to 83% (15/18 animals). On the other hand, according to the authors’ analyses, electrophysiological recordings (CAPs) and behavioral tests results did not differ significantly between the groups [55].

The first study on PNI treated with PEG in an animal model investigated the effects of the treatment depending on the time between injury and PEG application. When crushed sciatic nerve injury was treated using the PEG protocol within 30 min from the injury, the success rate, based on recovery of CAPs and muscle contractile force, reached 75% (6/8 animals). A 4 h delay after injury decreased the success rate to 66.7% (4/6 animals) [39]. This shows that the longer the time from PNI to PEG application, the lesser the benefits of PEG treatment. This raises the question of what the maximal reasonable time delay from PNI to the application of PEG treatment is. Fifteen years later, Bamba et al. tried to answer this question [56]. They evaluated nerve repair when PEG treatment was applied 1, 8, or 24 h after the sciatic nerve cut injury. At all time points and in each PEG-treated rat, PEG application restored postrepair CAPs conduction. Thus, the maximal time from injury to effective nerve fusion still remains undetermined.

Bamba et al. also incorporated PEG in human PNI therapy [64]. The enrolled patients met the following inclusion criteria: sharp nerve injury and time to surgical repair <12 h. Patients with avulsion injuries or injuries that occurred over 12 h prior to surgical intervention were excluded from the study. The outcomes were assessed clinically via static two-point discrimination (2PD) and the Semmes–Weinstein monofilament (SWM) test. The PEG-treated patients had significantly better Medical Research Council Classification (MRCC) scores at 1, 4, and 8 weeks after repair. The MRCC score was the only result in the manuscript provided for both the control group and the PEG-treated group. Unfortunately, the authors did not present the results of 2PD and SWM tests for the control group, which limits the interpretation of the outcomes of PEG treatment.

## 4. Perspectives and Future Directions

The nature of PEG-induced axonal fusion raises the question of its specificity in motor and sensory axons. Robinson and Madison investigated the impact of PEG treatment in a rat femoral nerve transection model [63]. Following 8 weeks of recovery, the terminal branches of the femoral nerve were exposed to retrograde fluorescent axonal dyes and later visualized in the spinal cord. In the PEG-treated group, the motor neurons exhibited no preference for the muscle pathway, whereas, in the control group, motor neurons preference for the muscle pathway was significant. The authors did not test motor function, thus the clinical significance of the reported reinnervation inaccuracy was undetermined in this model. Misdirection of regenerating axons occurs usually during the regeneration processes following nerve injury [65,66]. Neural plasticity in the central nervous system supports recovery and compensation after PNI. Therefore, the final regeneration outcome should be determined with respect to the recovery of nerve function.

PEG therapy in PNI treatment questions the dogma that axon severance irreversibly leads to Wallerian degeneration of the distal axon part. The results of animal studies are promising, though some issues still need further investigation before the introduction of PEG therapy in clinical practice. Among the functional measures of neuroregeneration after PEG treatment, only motor recovery was tested in the reported animal studies. Future experiments should incorporate sensory recovery analyses. A profound investigation into the recovery of all nerve functions (also autonomic) is crucial for considering translating PEG therapy to the clinics. Rapid motor recovery indicates restoration of orthodromic signal conduction through PEG-treated nerve lesion. Yet, no data on antidromic signal conduction has been published so far. Also, none of the published studies discusses the effects of PEG treatment on Schwann cells within the nerve injury region. Does reinnervation via PEG-mediated axonal fusion prevent Schwann cells death? Animal studies assessing changes in the postinjury, posttreatment survival rate of injured nerve Schwann cells should clarify this issue. The maximal delay in successful PEG therapy implementation has not yet been determined. Bamba et al. reported successful PEG-mediated fusion of the sciatic nerve 24 h after the initial nerve injury [56]. Establishment of an injury-to-intervention time limit on an animal model will set a foundation to plan a randomized clinical trial with objective tests for sensory and motor recovery evaluation in humans. A fact that is worth mentioning in case of a human randomized controlled trial (RCT) design is blinding the operator to the substance. Since PEG treatment in the described models incorporate intraoperative washes of the operation field with a characteristic set of solutions, proper blinding should be done.

## Figures and Tables

**Figure 1 ijms-20-01478-f001:**
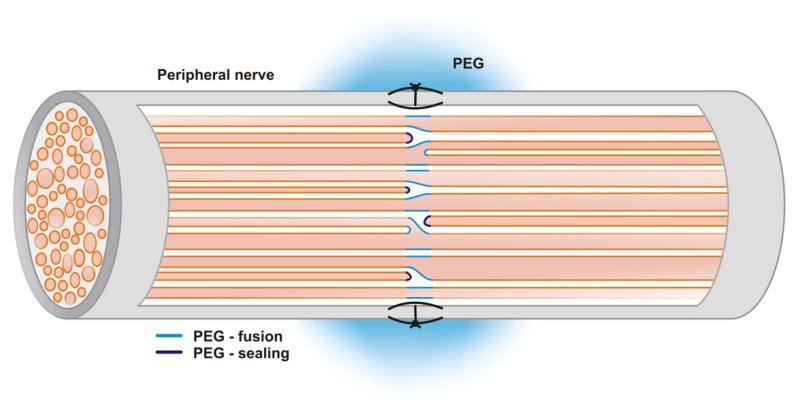
Mechanism of polyethylene glycol (PEG)-mediated fusion and sealing of axon ends. Polyethylene glycol application at the site of a nerve cut may result in either membrane fusion or membrane sealing.

**Figure 2 ijms-20-01478-f002:**
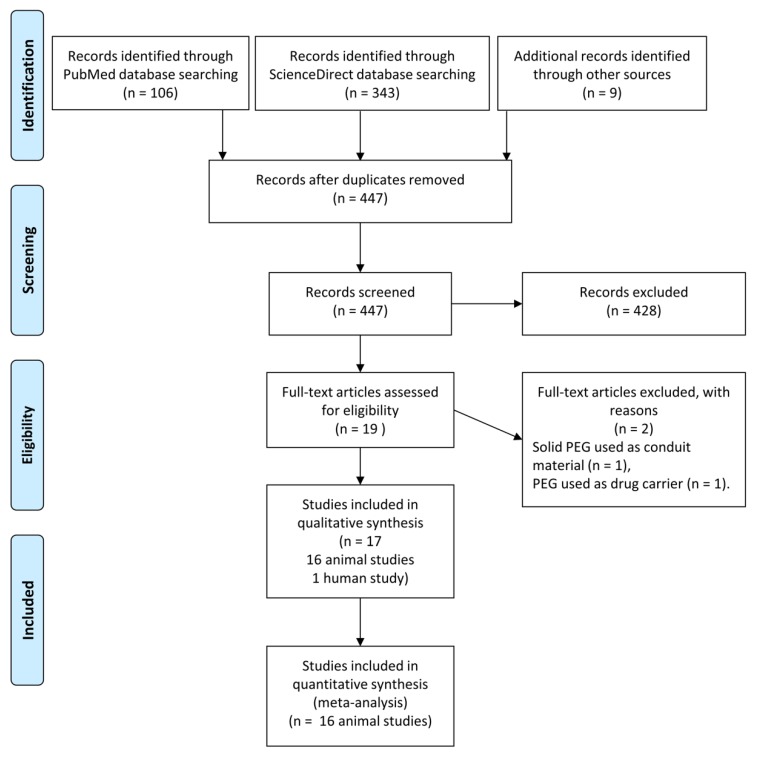
PRISMA flow diagram of the literature search for this systematic review.

**Table 1 ijms-20-01478-t001:** Animal studies of polyethylene glycol therapy in the treatment of peripheral nerve injuries. + present, − absent.

Animal Model	Type of Injury		Methods of Evaluation			PEG Therapy Superiority vs. Control Group	Reference
	Crush	Cut	Electrophysiological Recordings	Behavioral (Motor) Testing	Histological Analysis		
Guinea pig (sciatic nerve)	+	−	+	−	−	+	[39]
Rat (sciatic nerve)	+	−	+	−	−	+	[62]
Rat (sciatic nerve)	−	+	+	−	+	+	[48]
Rat (sciatic nerve)	+	+	+	+	−	+	[47]
Rat (sciatic nerve)	+	+	+	+	−	+	[52]
Rat (sciatic nerve)	−	+	+	+	+	+	[45]
Rat (sciatic nerve)	−	+	+	+	+	+	[54]
Rat (sciatic nerve)	−	+	+	+	+	+	[53]
Rat (sciatic nerve)	+	+	+	+	+	+	[46]
Rat (femoral nerve)	−	+	−	−	+	−	[63]
Rat (sciatic nerve)	−	+	+	+	+	+	[55]
Rat (sciatic nerve)	−	+	+	+	+	+	[56]
Rat (facial nerve)	−	+	+	−	+	+	[57]
Rat (facial nerve)	−	+	−	+	+	−	[60]
Rat (sciatic nerve)	−	+	+	+	+	+	[58]
Rat (sciatic nerve)	−	+	+	+	+	+	[59]

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
