# Peer review of "Polyethylene Glycol: The Future of Posttraumatic Nerve Repair? Systemic Review"

_ijms, 2019, doi:10.3390/ijms20061478_

Round 1

Reviewer 1 Report

 Minor spelling errors should be corrected. I would suggest that in the supplemental Table increment should be reduced in order to make data more easily readable or at least ensure that the same study is presented in the same page. 

Author Response

Dear Sirs or Madams,

We appreciate the reviewers’ comments and suggestions regarding our manuscript. In the following document, we have addressed all the issues raised by the reviewers. Also, we have revised our manuscript as suggested.

Reviewer 1:

Minor spelling errors should be corrected. I would suggest that in the supplemental Table increment should be reduced in order to make data more easily readable or at least ensure that the same study is presented in the same page. 

Author’s reply to reviewer report:

We have corrected the manuscript and reduced the supplemental table S1 as advised.

Reviewer 2 Report

The study of the effects of molecules for repairing the nerve injury, such as the polyethylene glycol, is without any doubt essential for searching new treatments that could give solution to this kind of trauma. For this reason, this systematic review presents a summary of relevant information. It is a well systematic review and the methodology is enough clear. However, some points should be clarified. 

Content points:

-In the manuscript authors mention the time of effectiveness of polyethylene glycol, but they do not mention information regarding the doses respect the models use and the effectiveness. There is lack of information about this question, what is mentioned in the studies cited?

-Did these studies in animal model and the clinical trial mention anything about the side-effects of polyethylene glycol, or possible inconvenient of using this kind of molecules? Authors should mention this, trying to explain some of the examples cited in the table, if it is possible.

-It is not enough commented the effects after the repair of the nerve at the levels commented (electrophysiological, histological, and behavioral). This information is summarized at table, but in the manuscript is not enough explained.

-Authors mentioned that the loss of Schwann cells is the main limitation in the axon regeneration process, but it is not clearly explained if at the same time that the polyethylene glycol acts as a fusogen, it also affects Schwann cells or myelin sheaths production.

-At perspective and future directions, it could be useful explain which are the specific future questions to be answer. What kind of information is needed, in which models, in order to could use this molecule in the future as a treatment in humans, since the majority of studies cited are in done in animal models. Authors can explain this point better.

Author Response

Dear Sirs or Madams,

We appreciate the reviewers’ comments and suggestions regarding our manuscript. In the following document, we have addressed all the issues raised by the reviewers. Also, we have revised our manuscript as suggested.

Reviewer 2:

The study of the effects of molecules for repairing the nerve injury, such as the polyethylene glycol, is without any doubt essential for searching new treatments that could give solution to this kind of trauma. For this reason, this systematic review presents a summary of relevant information. It is a well systematic review and the methodology is enough clear. However, some points should be clarified. 

Author’s reply to reviewer report:

We have addressed all the given comments.

Content points:

-In the manuscript authors mention the time of effectiveness of polyethylene glycol, but they do not mention information regarding the doses respect the models use and the effectiveness. There is lack of information about this question, what is mentioned in the studies cited?

Response: Polyethylene glycol (PEG) treatment consists of rinsing nerve injury site with PEG solution. What is crucial and variable in each summarized study is the composition of PEG solution (the molecular weight of used PEG and its concentration), time (how long PEG solution stays on nerve injury site before rinsing off) summarized in Table S1. Authors do not specify dose as a specific volume of solution, because a different volume is used in each used animal, depending on the size of the treated nerve. PEG dosing formulation recommends using as much solution as required to cover nerve injury site all around, leave the solution for a specified time depending on the used protocol. If provided by study authors, effectiveness of therapy is summarized in Table S1, column Results.

In order to clear that issue out, we have added a fragment in section 1.2. Mechanism of polyethylene glycol therapy of manuscript: „Worth highlighting is the fact, that protocols include no exact dose (volume) of PEG solution. The volume of solution needed to cover the whole treated nerve varies each time.”.

-Did these studies in animal model and the clinical trial mention anything about the side-effects of polyethylene glycol, or possible inconvenient of using this kind of molecules? Authors should mention this, trying to explain some of the examples cited in the table, if it is possible.

Response: In vivo studies included in the systemic review do not state / discuss any side-effects of PEG treatment. A review on PEG pharmaceutical applications by D’souza and Shegokar summarizes that: „short- and long-term toxicity studies, mutagenicity tests, teratogenic tests in rats, and human clinical trials have shown that PEG is safe via oral and non-oral routes.”1 However as every substance, if overdosed, can cause toxicity. PEG is often used as a drug carrier (PEGylated conjugates, surface-coating with PEG) and can lead to a formation of anti-PEG antibodies.1 Yet, single intraoperative exposition to PEG solution causes a relatively small risk of antibodies formation compared with chronic PEGylated drug administration.

1. D’souza, A. A., & Shegokar, R. (2016). Polyethylene glycol (PEG): a versatile polymer for pharmaceutical applications. Expert Opinion on Drug Delivery, 13(9), 1257–1275.doi:10.1080/17425247.2016.1182485 

We have added a fragment in section 1.2. Mechanism of polyethylene glycol therapy of manuscript: „From the clinical point of view, it is worth mentioning that PEG is considered safe to use in humans. Although no study included in the systemic review state any side-effects of PEG treatment, a review on its pharmaceutical applications by D’souza and Shegokar summarizes that toxicity, mutagenicity as well as teratogenic tests in rats, and human clinical trials have proven its safety via oral and non-oral routes.51 Still, PEG is often used as a drug carrier (PEGylated conjugates, surface-coating with PEG) which can lead to the formation of anti-PEG antibodies.51 However, a single intraoperative exposition to PEG solution causes a relatively small risk of antibodies formation compared with chronic PEGylated drug administration.”

-It is not enough commented the effects after the repair of the nerve at the levels commented (electrophysiological, histological, and behavioral). This information is summarized at table, but in the manuscript is not enough explained.

Response: We have added a fragment in section 3. Results of the manuscript: „In 14/16 experiments electrophysiological recordings were carried. Measured parameters included: CAPs, CMAPs (compound muscle action potentials) and muscle contraction force. Postrepair restoration of CAPs/CMAPs conduction through lesion site is an indicator of successful PEG-mediated axonal fusion. Based on the restoration of CAPs, the first 100% success rate of PEG treatment (n =32) was reported by Bittner et al. in a rat sciatic nerve cut injury model [52]. The behavioural (motor) testing is the most relevant part of neuroregeneration outcomes evaluation as it comes the closest to a clinical setting.  In most studies, that implemented motor recovery analysis, PEG-treated animals showed better performance than controls in short-term, which is consistent with a rapid nerve conduction recovery caused by PEG. In the long-term observation, this tendency persisted in all, except two experiments [47,53]. The histological analysis is the most objective method from those listed, as tissue harvesting unlike electrophysiological recordings and behavioural testing, is independent of animals generated disruptions. PEG-treated animals had a significantly higher number of axons in nerve parts distal to the injury site compared with controls (reported in 5 experiments, for details see Table S1) [45,53-56]. Also, axonal diameters in PEG-treated groups were significantly larger than respective controls [57-59].”.

-Authors mentioned that the loss of Schwann cells is the main limitation in the axon regeneration process, but it is not clearly explained if at the same time that the polyethylene glycol acts as a fusogen, it also affects Schwann cells or myelin sheaths production.

Response: Loss of Schwann cells after nerve injury is caused by denervation. Hypothetically, restoration of innervation, mediated by PEG axonal fusion, may inhibit Schwann cells death. However, we did not find any studies focusing on Schwann cells faith after PEG treatment, or how PEG treatment affects survival rate of Schwann cells. Indirectly, we can draw some conclusions out of histomorphometric nerve analyses e.g. myelin sheath area1,2, but these are carried usually at the end of long observation, leaving us unaware what happens to Schwann cells in early postinjury and posttreatment period. Also, most of the reported histomorphometric nerve analyses focus on axon features, neglecting myelin sheaths.

1. Riley, D.C.; Bittner, G.D.; Mikesh, M.; Cardwell, N.L.; Pollins, A.C.; Ghergherehchi, C.L.; Bhupanapadu Sunkesula, S.R.; Ha, T.N.; Hall, B.T.; Poon, A.D., et al. Polyethylene glycol-fused allografts produce rapid behavioral recovery after ablation of sciatic nerve segments. J Neurosci Res 2015, 93, 572-583, doi:10.1002/jnr.23514.

2. Mikesh, M.; Ghergherehchi, C.L.; Hastings, R.L.; Ali, A.; Rahesh, S.; Jagannath, K.; Sengelaub, D.R.; Trevino, R.C.; Jackson, D.M.; Bittner, G.D. Polyethylene glycol solutions rapidly restore and maintain axonal continuity, neuromuscular structures, and behaviors lost after sciatic nerve transections in female rats. J Neurosci Res 2018, 96, 1223-1242, doi:10.1002/jnr.24225.

We have added a fragment in section 4. Perspectives and future directions of manuscript: „Also, none of the published studies discusses the effects of PEG treatment on Schwann cells within the nerve injury region. Whether reinnervetion via PEG-mediated axonal fusion prevents Schwann cells death? Animal studies assessing changes in the postinjury, posttreatment survival rate of injured nerve Schwann cells should clarify this unknown.”

-At perspective and future directions, it could be useful explain which are the specific future questions to be answer. What kind of information is needed, in which models, in order to could use this molecule in the future as a treatment in humans, since the majority of studies cited are in done in animal models. Authors can explain this point better.

Response:

We have modified and added a fragment in section 4. Perspectives and future directions of manuscript:

„Polyethylene glycol therapy in PNI treatment questions the dogma that axon severance irreversibly leads to Wallerian degeneration of the distal axon part. The results of animal studies are promising though some issues still need further investigation before the introduction of PEG therapy in clinical practice. Out of functional measures of neuroregeneration after PEG treatment, only motor recovery was tested in reported animal studies. Future experiments should incorporate sensory recovery analyses. A profound investigation into the recovery of all nerve functions (also autonomic) is crucial for considering translating PEG therapy to clinical application. Rapid motor recovery indicates restoration of orthodromic signal conduction through PEG-treated nerve lesion. Yet, no data on antidromic signal conduction was published so far. Also, none of the published studies discusses the effects of PEG treatment on Schwann cells within the nerve injury region. Whether reinnervetion via PEG-mediated axonal fusion prevents Schwann cells death? Animal studies assessing changes in the postinjury, posttreatment survival rate of injured nerve Schwann cells should clarify this unknown. The maximal delay in successful PEG therapy implementation has not yet been determined. Bamba et al. reported successful PEG-mediated fusion of the sciatic nerve, 24 hours after the initial nerve injury [56]. Establishment of an injury-to-intervention time limit on an animal model will set a foundation to plan a randomized clinical trial with objective tests for sensory and motor recovery evaluation in human. A fact that is worth mentioning in case of a human RCT design is blinding the operator to the substance. Since PEG treatment in described models incorporate intraoperative washes of operation field with characteristic set of solutions, proper blinding should be done.”